# Fire History (1896–2013) in an *Abies religiosa* Forest in the Sierra Norte of Puebla, Mexico

Julián Cerano-Paredes [1], Dante A. Rodríguez-Trejo [2], José M. Iniguez [3,*], Rosalinda Cervantes-Martínez [1], José Villanueva-Díaz [1] and Osvaldo Franco-Ramos [4]

1. Centro Nacional de Investigación Disciplinaría en Relación Agua-Suelo-Planta-Atmósfera del Instituto Nacional de Investigaciones Forestales Agrícolas y Pecuarias, Km. 6.5 Margen Derecha del Canal Sacramento, Gómez Palacio 35140, Mexico; cerano.julian@gmail.com (J.C.-P.); cervantesmrosalinda@gmail.com (R.C.-M.); villanueva.jose@inifap.gob.mx (J.V.-D.)
2. División de Ciencias Forestales, Universidad Autónoma Chapingo, Texcoco 56230, Mexico; danteartuo@yahoo.com
3. USDA Forest Service, Rocky Mountain Research Station, 2500 S. Pine Knoll Drive, Flagstaff, AZ 86001, USA
4. Instituto de Geografía, Universidad Nacional Autónoma de México, Ciudad Universitaria, Coyoacán 04510, Mexico; ofranco@igg.unam.mx
* Correspondence: jose.m.iniguez@usda.gov

**Abstract:** The oyamel forests, as *Abies* dominated forests are commonly known as, register their largest distribution (95% of their population) along the Trans-Mexican Volcanic Belt (TMVB). Although efforts have been made to study these forests with various approaches, dendrochronology-based studies have been limited, particularly in pure *Abies* forests in this region. The objective of this study was to reconstruct fire regimes in an *Abies religiosa* forest in the Sierra Norte in the state of Puebla, Mexico. Within an area of 50-ha, we collected 40 fire-scar samples, which were processed and analyzed using dendrochronological techniques to identify 153 fire scars. The fire history was reconstructed for a period of 118 years (1896–2013), with low severity surface fires occurring mainly during in the spring (92.8%) and summer (7.2%). Over the past century, fires were frequent, with an mean fire interval (MFI) and Weibull median probability of (WMPI) of five years when considering all fire scars and less than 10 years for fires covering larger areas (fires recorded by ≥25% of samples). Extensive fires were synchronized with drought conditions based on Ring Width Indexes, Palmer Drought Severity Index (PDSI) and El Niño Southern Oscillation (ENSO). After 1983, we observed a change in fire frequencies attributed to regulated management. Longer fire intervals within the last several decades are likely leading to increased fuel accumulations and could potentially result in more severe fires in the future, threatening the sustainability of these forests. Based on our finding, we recommend management actions (silvicultural or prescribed fire) to reduce fuels and the risk of severe fires, particularly in the face of climatic changes.

**Keywords:** *Abies*; oyamel forest; fires; drought; ENSO; PDSI

## 1. Introduction

The *Abies* genus is widespread in boreal regions within the Northern Hemisphere but is limited to high elevation mountain areas in tropical regions [1,2]. In Mexico, this genus including 10 species [3] has an insular and irregular spatial distribution along the main mountain ranges of the country, with *Abies religiosa* (Kunth) Schltdl. et Cham. being the most abundant [4]. Oyamel forests, the common name for species of this genus in Mexico, cover 128,840-ha. *Abies* is also co-dominant over 28,877-ha for a total of 157,717-ha of the national territory [5]. The largest populations are located along the Trans-Mexican Volcanic Belt (TMVB) [4,6], which includes 95% of the total oyamel forest area [7].

Abies species in Mexico are susceptible to different disturbances, including land use changes [8], logging, overgrazing [1,6] and alteration of fire regimes [9,10]. These factors

have considerably reduced *Abies* populations [1,11] and six taxa are in some risk of local extinction category according to the Norma Oficial Mexicana (NOM-059-SEMARNAT-2010), which aims to identify species and populations of endangered flora and fauna [12]. Although these forests represent a small fraction of all forests in this region, the wildlife habitat, tourism, logging, water and other ecosystem services they provide are extremely important to the region. Similarly, these forests are threatened by increasing temperatures, droughts and wildfires associated with climate change [13]. Using various approaches, different aspects of these forests have been studied including; spatial distribution [14], diversity [6], structure [6,15–17], regeneration [18], germination [19], climate [20,21], ge-omorphology [22,23] and the effect of fires on regeneration and mortality [9,10]. Little, however, is known about the historical fire regime these forests are adapted to or how these regimes have changed as a result of increased human pressure.

In the last two decades, *Abies* forests in this region have experienced increasing patterns of wildfires, including patches of high severity fire. This is concerning because this species has relatively heavy seeds and is shade tolerant, suggesting that it is not adapted to high severity fires. Historical fire regimes based on dendrochronological techniques have been studied but only in mixed-conifer forests that include *Pinus-Abies-Pseudotsuga* in northern Mexico [24,25]. In central Mexico, where the largest *Abies* populations are concentrated, studies have mainly focused on the Monarch Butterfly Biosphere Reserve [26]. Therefore, this is the first study focused on fire regimes within pure *Abies* dominated forests in central Mexico. Given the threat of wildfire and the consequences of changing fire regimes in the western United States, understanding fire regimes in oyamel forests is a key element necessary for fire management programs and the long-term preservation and sustainability of these forests.

To develop sustainable fire management plans in any forest area, it is critical to first understand the fire history and how fire regimes have changed over time. This information is critical for understanding how forests and species have adapted to fire over long time periods [27]. The objective of this study was to reconstruct the historical fire regimes of an *A. religiosa* forest in the Sierra Norte de Puebla, Mexico, which had the highest number of fire incidences in the state [28]. More specifically, this study was designed to answer the following questions: (1) What were the fire regimes (fire frequency and seasonality) in these forests over the last century? (2) How were the fire regimes influenced by climate? and (3) Have the fire regimes changed in recent decades due to increased anthropogenic pressure?

## 2. Materials and Methods

### 2.1. Study Area

This study was conducted in the ejido Rinconada within the municipality of Chig-nahuapan, in the Northern Sierra of Puebla, between 19°41′56″–19°42′14″ N and 98°13′45″–98°14′09″ W (Figure 1). The forests of *A. religiosa* in this region are found at elevations between 2900 and 3100 m above sea level (asl). The climate is generally temperate with cool and rainy summers [29]. Based on recent records (1961–2000) at El Rosario weather station (2700 m above sea level) located three kilometers from the study area [30], the average monthly temperatures ranges from 10 °C minimum in the colder months (winter) to 16 °C in the warmer months (summer), with maximums between 20 °C and 24 °C. The average annual rainfall is 710 mm, with the highest proportion occurring in the summer and smallest proportion in the winter-spring seasons.

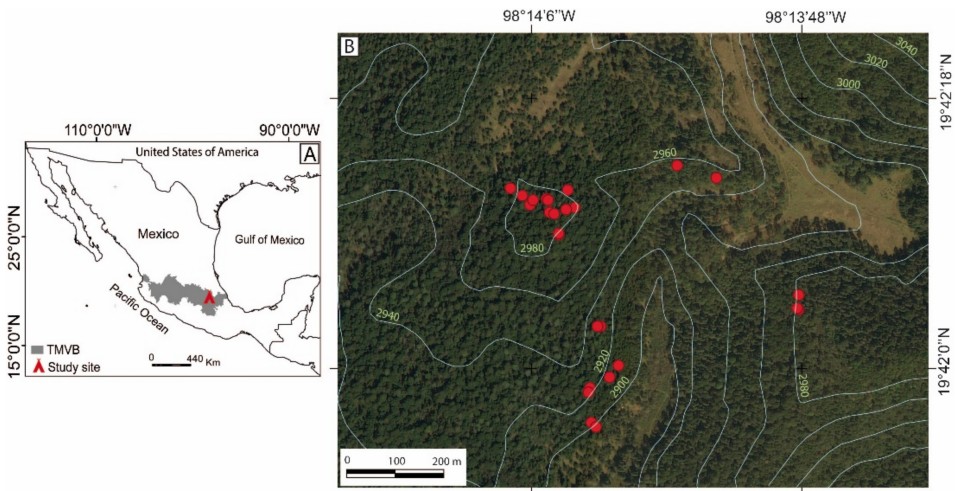

**Figure 1.** (**A**) Location of the study area in the Trans-Mexican Volcanic Belt (TMVB) and (**B**) distribution of sampling points (red dots) in the *Abies religiosa* forest in the Sierra Norte de Puebla, Mexico. The white lines are a 20-m interval contour line (altitude m a.s.l.).

### 2.2. Tree Sampling

Within the oyamel forest, we selected a sampling area of approximately 50 ha ranging between and selectively sampled 40 *A. religiosa* fire-scarred trees (Figure 2A,B). Using a chainsaw and standard methods [31], we collected 10 partial sections from living trees, where a small fire-scarred cross-section was removed while minimizing damage and preserving the trees' structural integration. The other 30 sections were collected from dead trees that were either standing (snags) or on the ground (logs) (Figure 2C). Fire scarred samples from dead trees were prioritized to minimize damage to living trees and to extract as long of a fire history record as possible [32]. Samples with the highest number of scars (fire records) were prioritized [31,33].

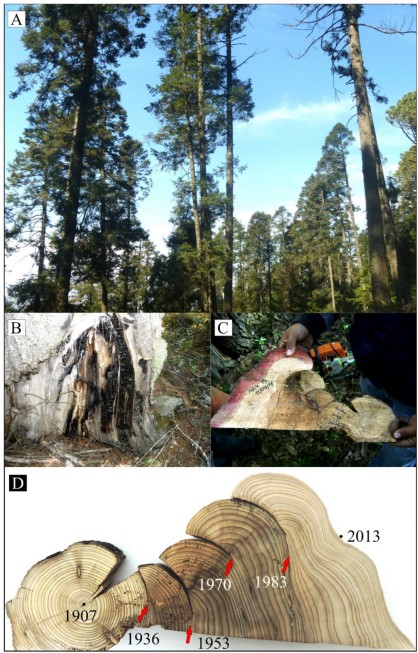

**Figure 2.** (**A**) Forest conditions within *Abies religiosa* forests in the Sierra Norte de Puebla, Mexico. (**B**) Base of a living tree with multiple visible fire scars. (**C**) Partial cross-section sampled from a living tree showing four fire scars. (**D**) Partial cross-section of a 107-years old *Abies religiosa* after it had been sanded and dated. The red arrows indicate fire-scars along with the calendar year of the fires (red and black numbers), as well as the pith and least recorded year.

### 2.3. Laboratory Methods

In the laboratory, all samples were sanded to a high polish using different sandpaper grains, ranging from 80 to 1200 grit, making it possible to determine individual tree rings under a microscope (Figure 2D). Each sample was then cross-dated using dendrochronological techniques to determine the exact calendar year of each fire-scar [34].

The season of fire occurrence was recorded based on the relative position of each fire scar within the annual ring (Figure 2D) and initially assigned to one of the following categories: D, dormant (between ring boundaries) EE, Early-early (first one third of earlywood), ME, middle early (second third of earlywood), LM, late earlywood (last third of earlywood), L, latewood [32,35]. The different categories were then grouped into two broader seasons: (1) spring (D + EE) and (2) summer (ME + LE + L) [36]. To analyze the seasonal weather condition and its influence on fire occurrence in both spring and summer, we graphed monthly averages for rain, temperature and thunderstorms activity using spatially explicit interpolated data based on nearby monitoring stations [37]. Likewise, the photosynthetic rate or stress level of vegetation (NDVI values) was analyzed throughout the year, as an additional variable allowing us to identify annual drought conditions and susceptibility of the vegetation to burning. NDVI values were extracted from the International Research Institute for Climate and Society (IRI) databases [38] and averaged on a monthly basis between 1981 and 2006.

### 2.4. Data Analysis

The fire history database was analyzed with the fire history program FHX2 version 3.2 [36]. The period of analysis began in 1921 when $\geq 10\%$ of the total number samples were recorded and ended in 2013 when samples were collected [39]. Fire intervals statistics were compared using three different filters: (1) All fires, included fire years that were recorded by at least one sample, (2) $\geq 10\%$ filter, included only years in which $\geq 10\%$ of the samples recorded a fire scar and (3) $\geq 25\%$ filter, included only fire years in which $\geq 25\%$ of the samples recorded a fire scar. Each of these fire-scar synchrony filters based on the proportion of samples that recorded a fire during a particular year is widely accepted in the fire history literature as a relative index of total area burned [40]. In particular, the $\geq 25\%$ filters was used here as an estimator of the widespread fires and contrasted to other filters that represent more localized fires and are therefore less synchronized in the fire scar records [41–43]. For each of the filters, we calculated the following descriptive statistics: mean fire interval (MFI), minimum and maximum interval, mean interval per sample and Weibull median probability interval (WMPI), which is the fire interval associated with 50% exceedance probability of a modeled Wuibull function fit to an empirical fire interval distribution. The latter is a central distribution measurement, used to model the asymmetric distribution of fire intervals and to express the intervals of recurrence in probabilistic terms [39,41].

### 2.5. Climate-Fire Relationship

To determine the relationship between climate and historical fire occurrence, climate variables were compared to fire occurrence using Superposed Epoch Analysis (SEA) in the FHX2 version 3.2 [36]. Fire occurrence data were then compared to three climate proxies including:

1. A local dendrocronological series based on *Pinus patula* [Schlechtendal and Chamisso] annual tree-rings from 1905–2011 [44].
2. The Winter Index (December-February) NIÑO 3 related to El Niño Southern Oscillation (ENSO) [45] and
3. The Palmer Drought Severity Index (PDSI) June-August (1900–2012) for the point closest to our study area, information extracted from the MXDA [46].

These three proxies represent climate conditions at three different scales. The annual tree-ring series represents site level climate conditions as recorded by *P patula*, which is a species sensitive to precipitation conditions. The PDSI is a more regional climate

index while the NIÑO 3 index is a sea-surface index indicating general climate conditions at continental scales. Each of the climatic variables was analyzed separately with the reconstructed fire history, comparing the climate conditions during the year of the fire, as well as conditions five years before and two years after it. To assess the statistical significance of SEA analysis, confidence intervals (95%, 99% and 99.9%) were calculated using the bootstrapped distribution of climate data with 1000 repetitions. In addition, the Atlas of Droughts for Mexico (MXDA) [46] was used to generate drought maps to visually analyze climatic conditions during the four most extensive fire years within the greater geographic region.

## 3. Results

A total of 153 fire scars were identified from 40 over a period of 118 years (1896–2013). The earliest recorded fire was in 1921 and the most recent in 2003. The most widespread fire year occurred in 1970, with 78% recording trees registering this event (Figure 3C). The fact that these fires were recorded on the tree-ring record as fire scars suggests that these fires were mainly low severity, surface or understory fires that had little or no effect on the mature tree canopy.

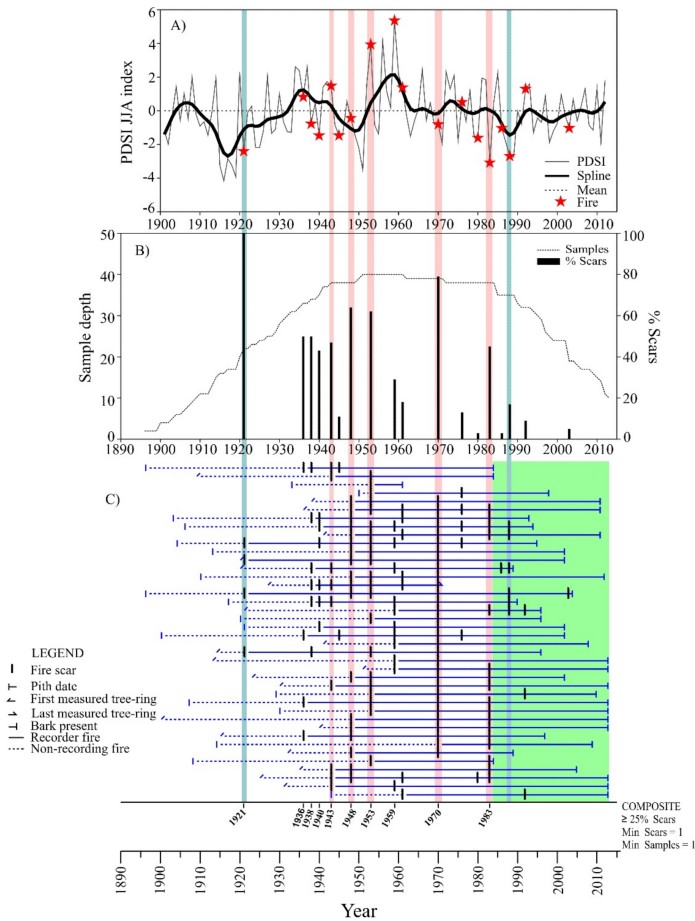

**Figure 3.** Fire history graph for an *Abies religiosa* forest in the Sierra Norte de Puebla, Mexico. (**A**) Palmer Drought Severity Index (PDSI) for the last century [46] and (**B**) percentage of fire-scarred trees per year, along with sample depth. (**C**) Timeline for each individual tree (horizontal lines), with vertical dashes indicating recorded fire-scars with fires recorded by ≥25% highlighted with dates below the graph. The green shading indicates a period of fire frequencies change after 1983 compared to prior period. The blue and pink bars indicate the drought condition during the most widespread fires.

### 3.1. Fire Frequency

In general, fires were frequent in these *A. religiosa* forests, with widespread fires (10 fires that scarred ≥25% of samples) being more common than localized fires (8 fires that scarred <25% of samples). The MFI and WMPI based on the all fires filter was less than 10 years, at 4.8 and 4.2, respectively (Table 1). Fires that were recorded in ≥10% of the samples were slightly more frequent, but the MFI and WMPI were also under 10 years (5.1 and 4.6 years, respectively). Fires that were recorded by ≥25% of the samples on average occurred at 7-year intervals (MFI and WMPI of 6.8 and 6.1 years, respectively). At the finer per sample scale, the MFI was longer at 13 years. The minimum fire-free interval was 2 years and the longest fire intervals was 15 years (Table 1). A change in fire frequency (Figure 3C, green area) was observed after 1983, with no widespread fires (fires recorded by ≥25% of the samples) in the past 20 years, a period that exceeds the maximum interval (15 years) previously recorded within these forests. In addition, there have been no synchronize fire events (fire recorded by two trees or more) in the study area since 1992.

**Table 1.** Fire interval descriptive statistics for an *Abies religiosa* forest.

| Period of Analysis | Category of Analysis | Number of Intervals | MFI | Min | Max | WMPI |
|---|---|---|---|---|---|---|
| 1921–2013 | All scars | 17 | 4.8 | 2 | 15 | 4.2 |
| | ≥10% scars | 13 | 5.1 | 2 | 15 | 4.6 |
| | ≥25% scars | 9 | 6.8 | 2 | 15 | 6.1 |

Note: MFI: Mean fire interval, Min: Minimum fire interval, Max: Maximum fire interval, WMPI: Weibull median probability interval.

### 3.2. Seasonality of Fires

Seasonality was determined for 90.2% of fire scars (Table 2), with most fires (92.8%) being recorded in the early earlywood (EE), with the remaining fire-scars being recorded either in the middle (5.1%) or late (2.1%) earlywood. No scars were recorded in the late wood and dormancy period. As a result, 92.8% of fires are considered to occur in spring and 7.2% in summer (Table 2), suggesting that most fires occurred in the seasonal dry period from March to June, given the Abies start the growing season in March in this region. Coincidently this is the time of the year with the highest number of lightning strikes and human caused fires in contemporary times, as well as the time period with the highest vegetation stress level due to a lack of moisture (Figure 4). It was not possible to determinate seasonality in 9.8% of the fire scars, because these involve an advanced rotting process of the wood (Table 2).

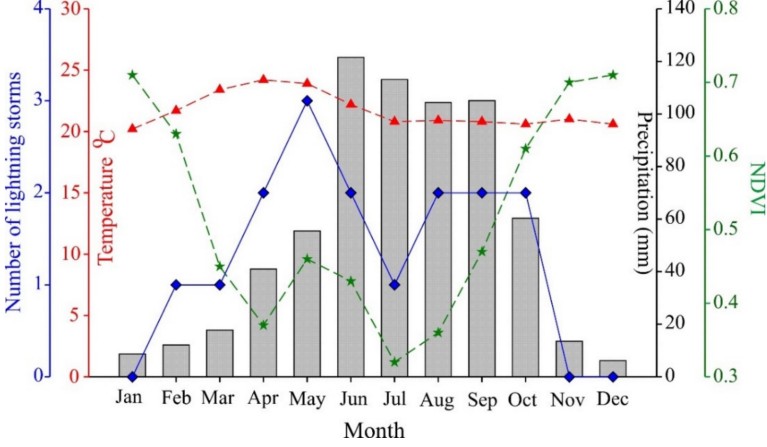

**Figure 4.** Monthly precipitation (bars) and temperature (red) recorded instrumentally within a weather station near the study area. The frequency of lightning storms is indicated in blue. Precipitation, temperature and lighting storm data are spatially explicit interpolated data based on nearby monitoring stations [37]. Green shows NDVI values, indicative of vegetation photosynthetic rates or vegetation stress levels (1981–2006) [38].

**Table 2.** Seasonality of fire scars in an *Abies religiosa* forest in central Mexico.

| Scars | Total | Determined Seasons | Undetermined Season | D | EE | ME | LE | L | Spring Fires [A] | Summer Fires [B] |
|---|---|---|---|---|---|---|---|---|---|---|
| Number | 153 | 138 | 15 | 0 | 128 | 7 | 3 | 0 | 128 | 10 |
| % | 100 | 90.2 | 9.8 | 0 | 92.8 | 5.1 | 2.1 | 0 | 92.8 | 7.2 |

Note: The definition of the seasonality of fires for this region of central Mexico was based on the methodology of [36]. [A] Dormancy + Early earlywood (D + EE). [B] Mid-earlywood + Late earlywood + Latewood (ME + LE + L).

### 3.3. Climate-Fire Relationship

Over the 118-year period of analysis, the SEA showed that the 10 most widespread fires ($\geq$25% of samples) reconstructed in the forest of *A. religiosa* coincided with a dry climate condition as indicated by below normal indexes of tree-ring width, and PDSI (Figure 5). In contrast, no fires were recorded during above-average moisture condition of tree ring-width, and PDSI (Figure 5). Similarly, no significant relationship was found one or two years immediately prior to fire events (Figure 5). Although no significant statistical relationship was found between fire occurrence and drought, most reconstructed fires for the period 1896–2013 occurred during years with below-average moisture conditions (11 out of 18 fires: 61%) (Figure 3A). Most widespread fires (recorded by $\geq$25% of samples) occurred during years with negative PDSI values (7 out of 10 fires, 70%) (Figure 3A), suggesting regional drought conditions in central Mexico (Figure 6). Likewise, these 10 larger fire events occurred during both ENSO years.

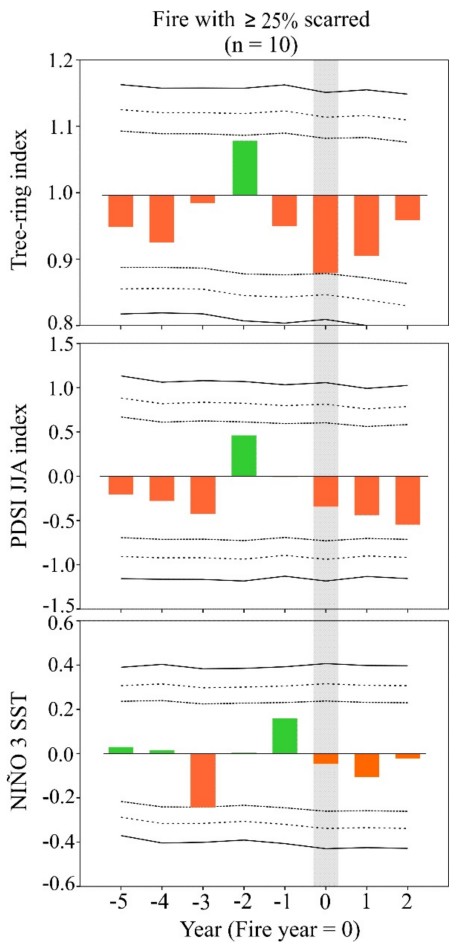

**Figure 5.** Superposed Epoch Analysis (SEA): Relationship between fire occurrences and regional climate (tree-ring index), PDSI (for the months June, July and August) and NIÑO 3 (Sea Surface Temperature)

indices. Fire years only includes years when $\geq$ 25% of samples were scarred. On the x axis, the year in which the fire occurred is year zero, with climate conditions on the *y*-axis including conditions five years prior to the fire (negative values) and two years after the fire (positive values). The top and bottom three lines within the graph represent the confidence intervals at 95%, 99% and 99.9%.

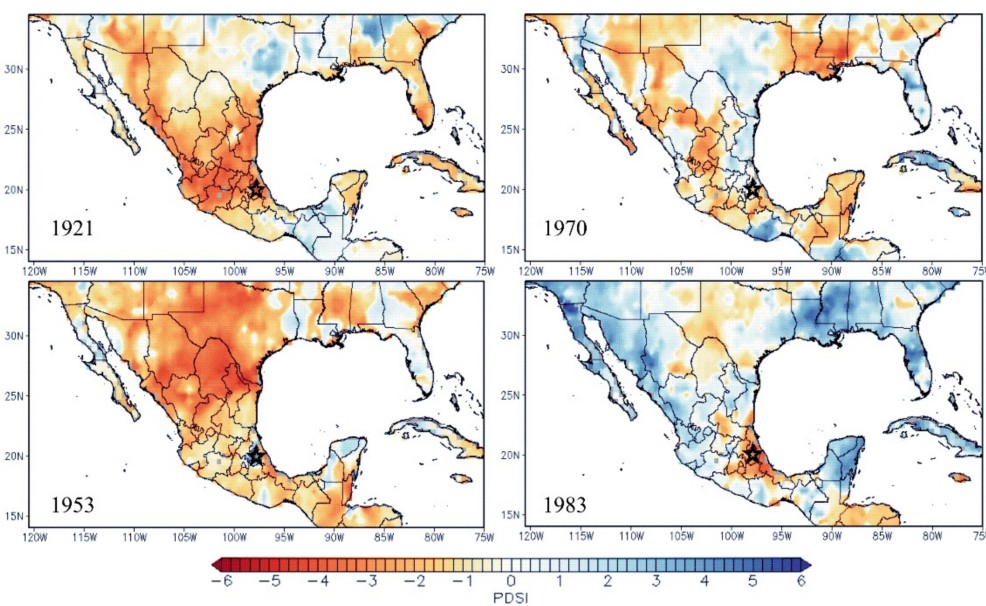

**Figure 6.** Geographic extent and severity of drought conditions in Mexico [46] during four widespread fire years. The black star symbol indicates the location of the study area.

## 4. Discussion

### 4.1. Fire Seasonality

Most of the fires in this *A. religiosa* forest occurred in spring (92.8%), which is consistent with the seasonality reported in *P. hartwegii* forest in this same region of the Sierra Norte de Puebla, where 92% of the fires were recorded in the beginning of the growing season [47]. A similarly pattern dominated by spring fires has also been reported in previous studies in *P. hartwegii* forests, in Pico de Orizaba [48], Sierra Norte de Puebla [47] and Cofre de Perote [49], all of which are located within the TMVB region. In the Sierra Norte de Puebla, spring fires in *A. religiosa* forests coincides with the period of least rainfall and highest temperatures of the year [37]. As a result, plant photosynthetic levels as measured by NDVI values were also low [38], indicating the highest level of annual vegetation stress (Figure 4). This inter-annual drought period also synchronized with the highest incidence of lightning storms (Figure 4) [37], a phenomenon that may have led to the natural onset of fires in these and in adjacent *P. hartwegii* forests. After the typically dry spring season, precipitation reinitiates limiting fire spread in *Abies* dominated forest. In the last three decades, however, this pattern has changed, with surface fires tending to occur early in the year, followed by an extended fire season that includes stand replacing (canopy fires) in April and May [50]. Similar seasonal patterns have been reported for the southwestern United States [51], as well as northern [52–54] and central Mexico [47,49,55].

### 4.2. Fire Frequencies

During the last century, our results show a pattern of frequent fires in the *A. religiosa* forests in the Sierra Norte de Puebla. When considering all fires, the mean fire interval was five years and remained below 10 years, even when only considering the most widespread events ($\geq$25% of the samples, Table 1). Of the 18 reconstructed fires at our study site between 1896 and 2013, 10 (56%) had high synchrony (Figure 3C), suggesting that these

were widespread fires [43]. These results address the first research question and are similar to results from other mixed conifer studies that include *Abies* species [51] in the southwestern United States and northern Mexico, which generally found that fire occurred once per decade before the current fire suppression period. Relatively short mean fire intervals of five years were also found in mixed conifer stands that included *Abies* in the states of Chihuahua [25], Durango [24], Michoacán and Mexico [26]. In addition, many of the fire years recorded within *A. religiosa* forests were synchronized with fire years in adjacent *P. hartwegii* forests located less than 5 km away [47]. In fact, 15 of the 18 fire years (83%) recorded in *A. religiosa* were synchronized with fire years for *P. hartwegii* forests. As mentioned by Rodríguez–Trejo [56], it is likely that, during wet years, oyamel forests serve as barriers to fire spreading between adjacent forest types, particularly pine forests. In addition, in ecotone areas that include pure pine and pine-oyamel forests, fire frequencies in *Abies* forests are similar to those found in pine forests [50].

Fire history studies conducted in central Mexico have documented uninterrupted fire frequencies up until the early part of the 21st century [26,48,49,56], while others have documented fire exclusion after 1970 [49]. Our results suggest the frequency of widespread fires changed after 1983 (Figure 3C). That is, during the period between 1983 and 2013 (30 years), fires were recorded in only four years (1986, 1988, 1992 and 2003), and on two of those years (1986 and 2003) fires were recorded by only one tree, suggesting highly localized events (Figure 3C). The decrease in fire frequency after 1983 is related to increased human influence in the local management. Local community records show increased logging activity after 1985, which implies greater canopy openings, more roads and more effective fire suppression [57]. This increase in anthropogenic activities coincided with changes in the tree-ring fire history record, particularly, changes in the fire frequency and an exclusion of widespread fires, which addresses our second research question.

The changing fire regimes we have documented here are an important factor in understanding current forests conditions and developing future conservation management plans to increase the sustainability of these forests. Longer fire free intervals are important because they often lead to an increased tree density [58], fuel accumulation (litter, grasses, herbs, shrub and woody material) including both horizontal and vertical fuel continuity [50, 59]. During the 1998 fire season stand replacing fires were observed within many *Abies* forests of central Mexico and in the nearby municipality of Ixtacamaxtitlán, in the state of Puebla [50], but not within our study area. Similar, interruption of the natural frequent fire regimes in the western forests of the United States has also led to increased fuel loads and subsequent stand replacing fires.

*4.3. Climate-Fire Relationships*

Interannual climate variability was not a statistically significantly factor in fire occurrence ($p > 0.05$), likely attributed to the low sample size in the SEA analysis. However, we did find that widespread fires ($\geq$25% of samples) were recorded during drought conditions, indicated by below-average climate index values (ring indexes, PDSI and NIÑO 3) (Figures 5 and 6). The influence of ocean-atmosphere phenomena such as ENSO (in its La Niña and El Niño phase) on climate variability and the occurrence of fires have been documented in both northern [52–54,59] and central Mexico [47,49]. This phenomenon is characterized by dipole conditions between northern and south-central Mexico. That is, warm phase (El Niño) ENSO events tend to result in drought conditions for the south-central and increase moisture in northern Mexico [53,60,61]. Conversely, cold phase (La Niña) ENSO events tend to produce drought conditions in the north and increased moisture in south-central Mexico. However, extreme La Niña events can produce drought conditions that extend into central Mexico, resulting in increased fire activity [49].

Of the 18 fire years reconstructed in our study area, 15 (83%) occurred in ENSO years, nine (60%) years were El Niño and six (40%) were La Niña years [62,63]. Of the 10 widespread fire years ($\geq$25% of samples) six (60%) occurred during the El Niño phase and four (40%) during the La Niña phase. Widespread fires in 1921, 1943 and 1970 were

recorded during extensive droughts conditions associated with La Niña (Figure 6). These were years of extreme La Niña droughts in northern Mexico that extended into central Mexico. Otherwise, the El Niño events in 1983 and 1998 that should have led to dry conditions for the southern part of Mexico also affected northern Mexico, where extensive high intensity fires were reconstructed at other sites during these two years [53]. These results are similar to climate-fire relationships described by Cerano–Paredes et al. [49], who documented extensive drought and fire events associated with both El Niño and La Niña, likely amplified by Pacific Decadal Oscillation (PDO). Although a fire was not recorded by tree-rings at this site in 1998, Rodríguez–Trejo [50] notes that the oyamel forests of the TMVB are more susceptible to stand replacing fires during El Niño years. Similarly, high severity fires have been documented by others within *Abies* forests in central Mexico during the El Niño years of 1998 and 2011 [10,50].

## 5. Conclusions

Of the 10 *Abies* species found in Mexico, *A. religiosa* is most commonly found within the country and provides a number of ecosystem services, as forests with this species tend to occur at the top of watersheds. Understanding the fire history of these fragile ecosystems is critical, given that forest fires are considered the most important threat to the sustainability of these forests under increasing temperatures related to climate change. In this study, we found that these forests have a history of frequent low severity fires throughout much of the 20th century. These low severity surface fires have helped maintained low tree densities and limited the accumulation of fuels. After 1983, however fire frequencies changed and widespread fires have been excluded from the study site since 1992.

Although complementary studies on forest structure are needed to determine current fuel loads, evidence of increased fuel loads is suggested by a recent patten of increasing larger and more severe fires in the region, as mentioned previously. That is, the lack of frequent fires has likely increased surface and ladder fuels, thereby increasing the probability of stand fires replacing crown fires, which these forests are not adapted to. Although our findings are based on a small 50 ha study area, similar fire frequency changes have likely occurred within most *Abies* forests in this region. This suggest that most *Abies* forests in this region could potentially be at risk of facing a high severity fire. Based on these findings, we recommend implementing forest management practices that decrease the vertical continuity of fuel and therefore the risk of surface and crown fires. These practices could include prescribed fire, thinning, clearings or any other fuel treatments that would reduce fuels loads. We also caution against the continued practice of fire suppression within these and other frequent-fire forests in Mexico, given the devastating consequences such management strategies have caused in similar forests across the western United States and elsewhere.

**Author Contributions:** J.C.-P., conceptualization, field collection, data analysis and original draft preparation; D.A.R.-T., draft preparation, review and editing; J.M.I., data analyses, draft review and editing; R.C.-M., dating fire samples and draft review; J.V.-D. and O.F.-R., draft review and editing. All authors have read and agreed to the published version of the manuscript.

**Funding:** This project was funded through the project "Reconstruction of the historical fire frequency and characterization of fuel loads in forest ecosystems in north-central Mexico", financed by the INIFAP fund.

**Institutional Review Board Statement:** Not applicable.

**Informed Consent Statement:** Not applicable.

**Data Availability Statement:** Data sharing not applicable.

**Acknowledgments:** We thank Vidal Guerra De La Cruz for site access and logistical support. We thank Gerardo Esquivel-Arriaga and numerous students for their assistance in field collections. This project was in part supported by the U.S. Department of Agriculture, Forest Service. We would also

like to thank the two anonymous reviewers for providing valuable feedback that greatly improved this paper.

**Conflicts of Interest:** The authors declare no conflict of interest.

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
