# Peer review of "Fire History (1896–2013) in an Abies religiosa Forest in the Sierra Norte of Puebla, Mexico"

_forests, doi:10.3390/f12060700_

Round 1

Reviewer 1 Report

The study of fire history is interesting and the observed chance in the fire regime can have important consequences in the futur. 

The manuscript can be shorten and many comments and suggestions are written directly into the manuscript.

Many main critic is the missing information of the fire regime (ground fire / canopy fire), the adaptation (or better the absence of adaptation) of Abies to canopy fire. The change in fire regime from ground to canopy fire would change the forest structure completly caused by fire suppression. The discussion could be oriented into this direction. This would give an interesting approach for this article.

The data of electric storms are interesting. Could you analyse the relationship between these events and fire occurrence by your? Il you found no relationship, you should omit this part in your results (figure 4).

I am not convinced that the NINO 3 SST analysis is necessary to show in figure 5. 

Several figure are not necessary and the analysis of LeNino are not very interesting in my mind. 

Author Response

The response to each of the comments from reviewer one are addressed in the comments section of the attached pdf file

Reviewer 2 Report

Comments and Suggestions for Authors

This is the review of the manuscript  (Manuscript ID: forests-1211942) from Julián Cerano-Paredes, Dante A. Rodríguez-Trejo, José M. Iniguez, Rosalinda Cervantes-Martínez, José Villanueva-Díaz and Osvaldo Franco-Ramos

Title: Fire history (1896-2013) in an Abies religiosa forest in the Sierra Norte of Puebla, Mexico

Journal: Forests, Submitted to section: Forest Ecology and Management

Authors analyzed the fire history in Abies religiosa forest in the Sierra Norte in Mexico.

 Overall, I found that the work is very interesting, the presented research material is  valuable.  I think, the manuscript can be interesting contribution for the readers of the journal. However, the manuscript needs improvement, especially figures. I have few comments and suggestions to you.

Below I list specific comments:

line 82 - 190 corrected

line 83 - add W for longitude

figure 1 write what the lines and unit mean (altitude m a.s.l.??), write that red dots - sampling points

line 121 - not P. but Pinus, why are fire-scars dating based on a chronology for a different species ??  describe how these chronologies are similar to each other

figure 2d - describe what the red arrows mean and whether the dates in white and black are different

line 150 - describe how WMPI was counted and add citation

line 167 - not P. but Pinus

line 199 - in 2002 or 2003 there is a fire-scar for the MAM16 tree ??

figure 3 - what does non-recording years mean ?? what mean recorder year ?? for me in 1921, 4 fire-scars are not for 4 trees but for 21 trees, then not 100% but 19%, I don't understand it, it needs to be changed or better explained, in caption (B) - no of reconstructed events, maybe reconctructed fires ??

Table 1 in the table is a description of Mfi and you write about MFI all the time, also give the unit for the table (years ??)

line 222 - is in March and early June, I think it should be from March to June

figure 4 - signature of the vertical axis No of electric storms

figure 5 - in the caption explain that PDSI is counted for JJA, and what SST means for NINO 3 ?? explain it

caption and eg line 339 - why is the tree-ring index included in the regional climate ?? I don't get it

figure 6 - change the star color to another, eg black, now illegible

figure 7 - what does 7-1), 7-2 and 7-3 should it be (1), (2) and (3) ??

line 339 negative climate index values ​​- PDSI and NONO 3 ok, but why tree index??

line 350 - unnecessary citations 63 and 64 - these are your results

line 377 - and the fire in 2002 or 2003 ??

Author Response

Response to the comments and suggestions by reviewer 2 are within the attached document

Round 2

Reviewer 1 Report

I did not see that my comments has been adresses by the authors.

Author Response

Please also see the attachment

We apologize for the oversight, but we initially thought that all of the reviewer 1 comments were all within the pdf documents and did not realize that reviewer 1 had also provided more comments separately.  In any case, here we address, to the best of our knowledge, both sets of comments from reviewer 1.  Below we have identified and bulleted both the first and second set of comment from reviewer 1 and provided a response that is preceded by Author’s response.  In addition, we have addressed all the specific comments from reviewer 1 within the pdf file, including comments from both the first and second set of comments.  Note that the reviewer may have to click on each of the yellow sticky notes to see our response.

Reviewer 1: first set of comments separate form from in the pdf file. 

  • The study of fire history is interesting and the observed chance in the fire regime can have important consequences in the futur. 

Authors response:  Thank you for the complement.  We agree that understanding the historical fire regime including how, when and why these have changed is important in regards to forest management.  This importance has been emphasized by increasing large and severe fires in the western US due to similar fire regime changes.

  • The manuscript can be shorten and many comments and suggestions are written directly into the manuscript.

Authors response:  Thank you for all 122 comments.  We greatly appreciate all the time and effort associated with such a detailed review which has greatly improved the paper.  We have responded to each of these comments within the attached pdf file. 

  • Many main critic is the missing information of the fire regime (ground fire / canopy fire), the adaptation (or better the absence of adaptation) of Abies to canopy fire. The change in fire regime from ground to canopy fire would change the forest structure completly caused by fire suppression. The discussion could be oriented into this direction. This would give an interesting approach for this article.

Authors response:  We agree that incorporating life history traits and adaptation strategies is important information that need to be provided to the reader in the introduction in order to understand why fire regimes and changes to fire regimes are important.  To that end, we have added such information to the introduction and elsewhere in the new version of the manuscript (attached).

  • The data of electric storms are interesting. Could you analyse the relationship between these events and fire occurrence by your? Il you found no relationship, you should omit this part in your results (figure 4).

Authors response:  The thunderstorm activity data is based on spatially explicit interpolated data, therefore because these are interpolated averages it is impossible to make direct comparisons with our re-constructed fire events.

  • I am not convinced that the NINO 3 SST analysis is necessary to show in figure 5. 

Authors response:  We believe including the NINO 3 SST analysis is necessary in figure 5 to explain the reason for the local and regional drought conditions.  That is, the tree-ring index is an indicator of local conditions, where as the PDSI is an indicator of regional conditions, however neither of these two explains why these conditions are occurring.  Therefore we need NINO 3 SST index to explain that the local and regional drought conditions are due to sea surface temperature changes that dictate global weather pattern.  Knowing this sort of relationship is useful in anticipating drought conditions in the future based on change in NINO 3 SST.

  • Several figure are not necessary and the analysis of LeNino are not very interesting in my mind. 

Authors response:  We have eliminated one of the figures and modified other according to the comment in the pdf file.  We have also eliminated the El Nino analysis from the results but have kept the discussion of El Nino is the discussion sections which includes more background information including the literature on this topic which is very extensive, suggesting other believe this is an interesting and important topic.  In this region in particular, El Nino years are dreaded because they tend to be associated with extensive drought and wildfire.  For example, the 1998 El Nino year was historic in terms for forest fires and associated damages.  Therefore anticipating such events is important in the preparation within regional and national governments.  

We received the following second set of comments from reviewer 1.  As before each of the comments from the reviewer is bulleted followed by the author's response to each comment.

  • I looked at my comments on the manuscript and the replay of the authors and their is non correlation. I still have no justification why the authors did not change in accordance to my suggestions.

Authors response:  We received 122 comment from reviewer 1 and did our best to respond to each comment within the short period of time (10 days).  However, we acknowledge there were a few (5) comment that we did not respond to.  In this second set of review we have addressed all comment as best of our knowledge. 

  • The Most of my comments (for example to exclude the analysis of Le Nino, and the simplification of only spring (earlywood fire scars) and summer (latewood fire scars) fire events was not addressed at all.

Authors response:  These two comments are addressed within the pdf file near lines 227 and 250.

  • Thus, I really need to get explanation by the author ignore so many of my comments.

Authors response:  As mentioned previously there were 122 comment and we apologize for not noticing the comments provide separately from the pdf file.  We hope that this document along with the pdf file and associated comments is satisfactory.

  • I still do not feel to accept this version.

Authors response:  We feel this is unfortunate and hope we have now addressed all comments in a manor what would make this manuscript acceptable.

  • Also, the reply letter should indicate if the lines referred to the initial version of the manuscript to the new version. Maybe, I am the first reviewer?"

Authors response:  The comment we received from reviewer 1 were as sticky notes within a pdf file which make it difficult/impossible to determine the line number.  As a result, we decided to provide our Author’s response below each of the original reviewer’s comment.  Also as mentioned before we did not notice the comments provided by reviewer 1 separate from the pdf file, however we have not responded to all comments.  

Please also see the attachment
